# CAT-SAC: Soft Actor-Critic with Curiosity-Aware Entropy Temperature

## Abstract

The trade-off between exploration and exploitation has long been a crucial issue in reinforcement learning (RL). Most of the existing RL methods handle this problem by adding action noise to the policies, such as the Soft Actor-Critic (SAC) (Haarnoja et al., 2018a;b) that introduces an entropy temperature for maximizing both the external value and the entropy of the policy. However, this temperature is applied indiscriminately to all different environment states, undermining the potential of exploration. In this paper, we argue that the agent should explore more in an unfamiliar state, while less in a familiar state, so as to understand the environment more efficiently. To this purpose, we propose **C**uriosity-**A**ware entropy **T**emperature for SAC (CAT-SAC), which utilizes the curiosity mechanism in developing an instance-level entropy temperature. CAT-SAC uses the state prediction error to model curiosity because an unfamiliar state generally has a large prediction error. The curiosity is added to the target entropy to increase the entropy temperature for unfamiliar states and decrease the target entropy for familiar states. By tuning the entropy specifically and adaptively, CAT-SAC is encouraged to explore when its curiosity is large, otherwise, it is encouraged to exploit. Experimental results on the difficult MuJoCo benchmark testify that the proposed CAT-SAC significantly improves the sample efficiency, outperforming the advanced model-based / model-free RL baselines.

## 1 Introduction

Deep reinforcement learning (RL), with high-capacity deep neural networks (DNNs), has been applied to solve various complex decision-making problems, including video games (Mnih et al., 2015; Vinyals et al., 2019), chess games (Silver et al., 2017; 2018) as well as robotic manipulation (Kalashnikov et al., 2018). However, it is notorious for its high sample inefficiency (Kaiser et al., 2019). Even when solving a simple task, RL needs substantial interaction data to improve the policy. One of the major obstacles for achieving sample-efficiency is the difficulty of balancing exploration and exploitation. The RL agent needs to explore the environment to collect useful information as well as exploit the acquired knowledge to improve its policy. Most of the existing works either use intrinsic rewards (e.g., count-based bonuses (Bellemare et al., 2016) and state prediction error (Pathak et al., 2017; Burda et al., 2018b)) to strengthen exploration, or augment the action value with the entropy to control the proportion of exploration dynamically (Haarnoja et al., 2017; Ziebart et al., 2008). Among these approaches, Soft Actor-Critic (SAC) (Haarnoja et al., 2018a;b) achieves the superior performance on MuJoCo (Todorov et al., 2012), an OpenAI gym (Brockman et al., 2016) benchmark with complex continuous control tasks.

Specifically, SAC aims to maximize expected value while also maximizing the entropy:

$$\pi^* = \arg\max_{\pi} \sum_{t=0}^{\infty} \mathbb{E}_{s_t, a_t \sim \rho_\pi}[r_t + \alpha \mathcal{H}(\pi(\cdot|s_t))], \tag{1}$$

where $r_t$ is the environment reward at timestep $t$, $s_t$ is the state, $a_t$ is the action, $\rho_\pi$ is the distribution of trajectory w.r.t. policy $\pi$, $\mathcal{H}$ is the entropy of the policy and $\alpha$ is the entropy temperature that weights the importance of the entropy term versus the external reward. The entropy temperature is critical in SAC since different values determine diverse patterns of agent behavior. Small $\alpha$ may lead the agent to over-optimize the state-action value and develop a greedy policy. Due to the lack of

exploration, the agent with a small $\alpha$ is likely to get stuck in a local optimal. On the other hand, over-optimize the entropy with large $\alpha$ causes the agent to act nearly uniformly and therefore hampers the ability to exploit the environment. To choose a reasonable temperature automatically, Haarnoja et al. (2018b) introduce a target entropy term and extends SAC to a constrained optimization problem, which performs empirically well on MuJoCo tasks (Brockman et al., 2016).

However, the target entropy is applied to all transitions equally during training, which neglects the particularity of different states – some states require less exploration while others need more (Tokic, 2010). The particularity of different states for exploration has been discovered for a long time. One of the important particularity is the internal curiosity w.r.t. the environment states. Intuitively, when playing a new game, humans make many different attempts at the beginning, and after they understand the basic logic of the game, they will not spend extra time trying the things they already know. This implies that a better exploration strategy is to explore when the agent is in unfamiliar states and exploit when it is in the familiar states. Therefore, optimizing the policy with a global target entropy without considering the particularity of different states may instead hamper the exploration of the agent. Researchers use internal curiosity as an auxiliary reward to encourage the exploration of the agent (Bellemare et al., 2016; Tang et al., 2017; Ostrovski et al., 2017). Among these methods, Burda et al. (2018b) propose Random Network Distillation (RND) (Burda et al., 2018b) to remedy the 'noisy TV' problem (Pathak et al., 2017)[1], which predicts the output of a fixed random network and demonstrates the excellent exploration ability on games with image input. Although Haarnoja et al. (2018b) have also considered the association between the curiosity and the entropy, they do not apply any relevant guidance to their proposed strategy of automatically tuning entropy temperature.

In order to dynamically adjust the exploration-exploitation strategy regarding the curiosity about states, we make the first attempt to introduce the curiosity to the target entropy term of SAC so that SAC can actively enlarge its entropy at unfamiliar states while reducing its entropy at familiar states. We named our method as CAT-SAC – SAC with **C**uriosity-**A**ware entropy **T**emperature, by seamlessly introducing the curiosity to augment the target entropy term. Specifically, CAT-SAC first augments the target entropy term by adding the zero mean curiosity. By this means, the agent is encouraged to explore more in an unfamiliar state as its target entropy is large (with positive curiosity) and to explore less at a familiar state as its target entropy is small (with negative curiosity). Then, CAT-SAC adopts an instance-level entropy temperature $\alpha(\text{state})$ to replace the original global entropy temperature so as to learn different entropy temperatures for different states. The instance-level entropy temperature is curiosity-aware as it is supervised by the curiosity augmented target entropy. As for the curiosity model, current advanced methods e.g. RND (Burda et al., 2018b) may fail to model the curiosity for feature input (e.g., position, velocity). One of the major reasons is that different states with dramatic visual difference may have a little difference in feature so that RND has difficulty in distinguishing different state with feature inputs and fails to model the curiosity. To cope with this problem, we propose a novel curiosity model X-RND, which first synthesizes 'unvisited' feature states from the collected data by blending two visited states with random weight entry by entry. With the visited and the 'unvisited' states, X-RND learns to separate the 'unvisited' from the visited by contrastive self-supervised learning (He et al., 2020). By doing so, X-RND successfully prevents the curiosity about the 'unvisited' states from the influence of the visited states.

In summary, the main contributions of this paper are as follows: We propose a novel CAT-SAC model, which enables the agent to better trade-off the exploration versus exploitation according to different curiosity about states; To model curiosity for feature inputs, we propose a new curiosity model, X-RND, optimized by contrastive self-supervised learning. Experimental results testify that the proposed method significantly improves the sample-efficiency on complex and difficult continuous control tasks of the MuJoCo benchmark against the state-of-art model-based / model-free methods.

## 2    RELATED WORK

Exploration is an essential issue for effective reinforcement learning (Kakade & Langford, 2002), and the problem of exploration has been widely studied (Wunder et al., 2010; Haarnoja et al., 2018a; Pathak et al., 2017; Lee et al., 2020; Zhang et al., 2020; Sekar et al., 2020; Stadie et al.,

---

[1] When predicting the dynamics, the curiosity model is prone to produce a large error w.r.t noisy observation, resulting in a large curiosity.

2018; Nachum et al., 2019). The main researches on exploration include noise-based exploration (Wunder et al., 2010; Tokic, 2010; Mnih et al., 2015; Lillicrap et al., 2015), maximum entropy of policy exploration (Haarnoja et al., 2017; 2018a) and intrinsic bonus exploration (Pathak et al., 2017; Burda et al., 2018a; Achiam & Sastry, 2017; Burda et al., 2018b). Noise based exploration aims to balance the ratio of exploration/exploitation by introducing a random policy noise. The most relevant to CAT-SAC is the maximum entropy exploration and the intrinsic bonus.

**Maximum Entropy RL.** Maximum entropy RL optimizes policies by maximizing the expected return and the expected entropy of the policy simultaneously (Haarnoja et al., 2017) (Haarnoja et al., 2018a). This framework has been used in many fields, such as reverse reinforcement learning (Ziebart et al., 2008) and optimal control (Roy et al., 2013). Recently, the maximum entropy framework has been successfully incorporated into Q-learning and policy gradient methods to improve sample efficiency, such as soft Q-learning algorithm (Haarnoja et al., 2017) and soft actor-critic (SAC) algorithm (Haarnoja et al., 2018a; 2019). Soft Q-learning learns stochastic energy-based policies with approximate inference via stein variational gradient descent (SVGD) (Liu & Wang, 2016). However, approximate inference procedures become complex in continuous action spaces. Haarnoja et al. (2018a) present a convergence proof for policy iteration in the maximum entropy framework, known as soft actor-critic (SAC), which avoids the complexity and potential instability associated with the approximate inference of soft Q-learning. Haarnoja et al. (2019) further propose an extension to the soft actor-critic algorithm that the temperature parameter can be optimized adaptively.

**Intrinsic Bonus.** Intrinsic bonus exploration methods encourage the agent to explore by introducing an intrinsic bonus as an auxiliary reward. One of the common intrinsic bonuses is defined based on visit-count that indicates how many times the agent performed action $a$ at state $s$ (Bellemare et al., 2016; Tang et al., 2017; Ostrovski et al., 2017; Fox et al., 2018). Another class of intrinsic bonus is calculated according to errors of predicting dynamics (Pathak et al., 2017; Burda et al., 2018a; Achiam & Sastry, 2017; Houthooft et al., 2016). They first learn forward dynamics to predict the next observation given the current observation and agent's action. The prediction error represents the agent's familiarity about the state. These approaches focus on maximizing the prediction error as an intrinsic reinforcement learning objective. However, maximizing such prediction errors of forwarding dynamics will tend to cause the agent to be attracted to stochastic transitions (Burda et al., 2018b), e.g., the "noisy TV". Burda et al. (2018b) propose random network distillation (RND) bonus to handle this undesirable stochasticity. RND predicts the output of a fixed randomly initialized neural network based on the current observation. Therefore, the predicted output is deterministic.

**Undirected Exploration Strategy.** Even though intrinsic bonus methods have obtained a significant improvement on (semi-)sparse tasks, they also introduce exploration biases to the policy, which might cause the policy to improve in the opposite direction against the original reward function. Such a "misdirected" problem becomes even severe on dense-reward tasks since the original policy could have performed much better. Therefore, there are also plenty undirected exploration methods, such as $\epsilon$-greedy (Sutton & Barto, 2018) and posterior sampling (Osband et al., 2013; Osband & Van Roy, 2015; Kurutach et al., 2018; Wang & Ba, 2019).These methods focus on adjusting the variance of policy to explore better. As studied in (Tokic, 2010), increasing policy variance at unfamiliar states cause the policy to explore better. Similar ideas are also be observed in bootstrapping methods (Osband et al., 2016; 2018) and upper-bound-confidence exploration methods (Ciosek et al., 2019; Lee et al., 2020), which tend to have larger action variance at uncertainty states. In this paper, we propose to introduce the unfamiliarity of states to automatically tune the entropy temperature to improve the exploration of SAC.

## 3 Preliminaries

In this section, we introduce the background and annotations of reinforcement learning and the two key ingredients of our method, i.e., SAC (Haarnoja et al., 2018a;b) and RND (Burda et al., 2018b).

### 3.1 Reinforcement Learning

We consider a standard RL framework where an agent interacts with an environment according to the observation in discrete time. Formally, the agent observes a state $s_t$ (e.g., position, velocity)

from the environment at each time step $t$ and then chooses an action $a_t$ according to its policy $\pi$. The environment returns a reward $r_t$ and the next state $s_{t+1} \sim p_e(s_t, a_t)$, where $p_e$ is the environment transition mapping. The return of a trajectory from timestep $t$ is $\eta_t = \sum_{k=0}^{\infty} \gamma^k r_{t+k}$, where $y \in [0, 1)$ is the discount factor. RL aims to optimize the policy to maximize the expected return from each state $s_t$.

## 3.2 SOFT ACTOR-CRITIC

SAC (Haarnoja et al., 2018a) is an off-policy actor-critic method based on the maximum entropy RL framework (Ziebart et al., 2008), which aims to maximize both the return and the policy entropy, as described in Equ. (1). It uses a neural network with parameter $\theta$ for Q-value $Q_\theta$, and uses anther neural network parameterized by $\phi$ for the policy $\pi_\phi$. To update these parameters, SAC alternates between a soft policy evaluation and a soft policy improvement. At the soft policy evaluation step, $\theta$ is updated by minimizing the following cost function:

$$\mathcal{L}_{\text{critic}}(\theta) = \mathbb{E}_{s_t,a_t \sim \mathcal{D}}[||Q_\theta(s_t, a_t) - Q^{\text{target}}(s_t, a_t)||_2^2], \tag{2}$$

$$\text{where } Q^{\text{target}}(s_t, a_t) = r_t + \gamma \mathbb{E}_{s_{t+1} \sim p_e, a_{t+1} \sim \pi_\phi}[\hat{Q}(s_{t+1}, a_{t+1}) - \alpha \log(\pi_\phi(a_{t+1}|s_{t+1}))], \tag{3}$$

where $\mathcal{D}$ is an experience replay buffer, $\alpha$ is the entropy temperature, and $\hat{Q}$ is the target Q function whose parameters are periodically updated from the learned $Q_\theta$. At the soft policy improvement step, the policy $\pi$ with its parameters $\phi$ is updated by minimizing the following objective:

$$\mathcal{L}_{\text{actor}}(\phi) = \mathbb{E}_{s_t \sim \mathcal{D}, a_t \sim \pi_\phi}[\alpha \log \pi_\phi(a_t|s_t) - Q_\theta(s_t, a_t)]. \tag{4}$$

Here, the policy models a Gaussian with mean and variance to handle continuous control problems. In order to automatically tune $\alpha$, Haarnoja et al. (2018b) introduces a constraint target entropy $\tilde{\mathcal{H}}$, so that $\alpha$ is learned to meet the constraints during the whole training process:

$$\mathcal{L}(\alpha) = \mathbb{E}_{s_t \sim \mathcal{D}, a_t \sim \pi_\phi}[-\alpha\big(\log \pi(a_t|s_t) + \tilde{\mathcal{H}}\big)]. \tag{5}$$

## 3.3 RANDOM DISTILLATION NETWORK

Random Network Distillation (RND) (Burda et al., 2018b), a prediction-based method for encouraging exploration through curiosity, which manages to exceed average human performance on Montezuma's Revenge (Badia et al., 2020). The RND exploration bonus is defined as the state prediction error of a neural $f_\omega(s_t)$ predicting features of the state given by a fixed randomly initialized neural network $\hat{f}(s_t)$. The exploration bonus and the loss function are:

$$c_\omega(s_t) = ||f_\omega(s_t) - \hat{f}(s_t)||_2^2 \text{ and } \mathcal{L}(\omega) = \mathbb{E}_{s_t \sim \mathcal{D}}[c_\omega(s_t)]. \tag{6}$$

In unfamiliar states, it's hard for $f_\omega$ to predict the output of $\hat{f}$, and hence the prediction error $c(s_t)$ is high. By measuring how hard it is to predict the output of $\hat{f}$ on these states, RND incentivizes visiting unfamiliar states.

# 4 CAT-SAC

In this section, we present CAT-SAC : **C**uriosity-**A**ware entropy **T**emperature for Soft Actor-Critic. In the design of CAT-SAC, we integrate three key ingredients, i.e., the curiosity augmented target entropy, the instance-level entropy temperature, and the curiosity model X-RND for feature input, into a single framework.

## 4.1 CURIOSITY AUGMENTED TARGET ENTROPY

Haarnoja et al. (2018b) reformulates SAC from the maximum entropy RL framework to a constraint satisfaction problem by introducing the target entropy to tune the entropy temperature automatically. However, applying the target entropy to all transitions neglects the particularity of different states, i.e., an unfamiliar state often requires more exploration. To enable the agent to increase the entropy

when it's in an unfamiliar state, we inject the curiosity to the target entropy. Specifically, we first normalize the curiosity $c(s)$ w.r.t state $s$, and then add it to the original target entropy $\tilde{\mathcal{H}}$:

$$h(s) = \tilde{\mathcal{H}} + \frac{c(s) - \mu}{\sigma}, \tag{7}$$

where $\mu$ and $\sigma$ are the running mean and running standard deviation of $c(s)$ respectively. From the Equ.(7), the expectation of the proposed curiosity augmented target entropy is $\mathbb{E}_s[h(s)] = \tilde{\mathcal{H}}$, therefore the new target entropy is consistent with the original one in expectation. In this manner, the target entropy in an unfamiliar state is pushed above the expectation and thus the agent is encouraged to explore more in that state; on the contrary, the target entropy in the familiar state is pressed down to regulate the behavior of the agent to exploit efficiently.

## 4.2 Instance-level Entropy Temperature

The entropy temperature is the most crucial component in either SAC or CAT-SAC, as it is the bridge connecting the target entropy and the policy. Therefore, to really cast different target entropy to the policy at different states, we need to turn the original global entropy temperature $\alpha$ into an instance-level entropy temperature $\alpha_\delta(s)$, whose target is:

$$\delta^* = \operatorname*{argmin}_\delta \mathbb{E}_{s_t \sim \mathcal{D}, a_t \sim \pi_\phi}[-\alpha_\delta(s_t)\big(\log \pi_\phi(a_t|s_t) + h(s_t)\big)]. \tag{8}$$

From Equ.(8), we can observe that if $\alpha$ is state-independent, then the expectation term in Equ.(8) will be equivalent to the original of Equ.(5). In this case, the effect of the curiosity augmented target entropy will not be reflected in the policy as its expectation is equal to the original target entropy.

However, assigning different entropy temperature for each individual state independently makes the optimization of Equ.(8) trivial: the entropy temperature $\alpha_\delta(s_t)$ become either extremely high or extremely low according to the sign of $\log \pi_\phi(a_t|s_t) + h(s_t)$. It further over-weights $\log \pi_\phi(a_t|s_t)$ of Equ.(4), which influences back the optimization of Equ.(8). The consequence of these optimization processes is that $-\log \pi_\phi(a_t|s_t)$ is optimized to approach $h(s_t)$ rapidly. As mentioned by Haarnoja et al. (2018b), forcing the entropy to be a target entropy will hamper the flexibility of the policy. To prevent the $\alpha_\delta(s_t)$ from doing so, instead of $s_t$, we use the curiosity $c(s_t)$ as the context of the entropy temperature, which projects different states to compact values so as to cluster those states with the same level of unfamiliarity to share the same entropy temperature:

$$\alpha_\delta(s_t) = g_\delta(c(s_t)), \tag{9}$$

where $g_\delta$ a linear layer with parameter $\delta$. More details about the $g_\delta$ is presented in Appendix. To further alleviate the problem mentioned above, we project the zero-mean $c(s_t)$ into its closest integer to reduce the space of the entropy temperature. One can treat our method as a stratified SAC regarding different discrete curiosity levels.

## 4.3 X-RND Based on Self-supervised Contrastive Learning

In our framework, the curiosity values of familiar states should be relatively smaller than those of unfamiliar states. Burda et al. (2018b) have demonstrated that RND works well in modeling curiosity for hard exploration tasks with image input. However, we found that RND works poorly with feature input. Especially for the features of unvisited states that are close to the visited states', RND is likely to produce low prediction errors. Therefore, the curiosity model based on state prediction error like RND, besides minimizing the prediction error Equ.(6) of the visited states, may need extra supervision to learn how to maintain the prediction error of the unvisited states to a certain extent. To this end, we develop X-RND, which includes an additional contrastive loss (He et al., 2020; Schmidhuber, 2015) to push the curiosity values of potential unseen states $s'$ above a margin value. The loss function of X-RND is formulated as:

$$\mathcal{L}(\omega) = \mathbb{E}_{s_t \sim \mathcal{D}}\big[c_\omega(s) + \beta \max(m - c_\omega(s'), 0)\big], \tag{10}$$

where $m$ is the curiosity margin for unvisited states and $\beta$ is the hyperparameter that weighs the importance of the contrastive loss term. In order to obtain the unvisited states $s'$, X-RND constructs the potential unvisited states from the visited states. Specifically, X-RND randomly samples a set of

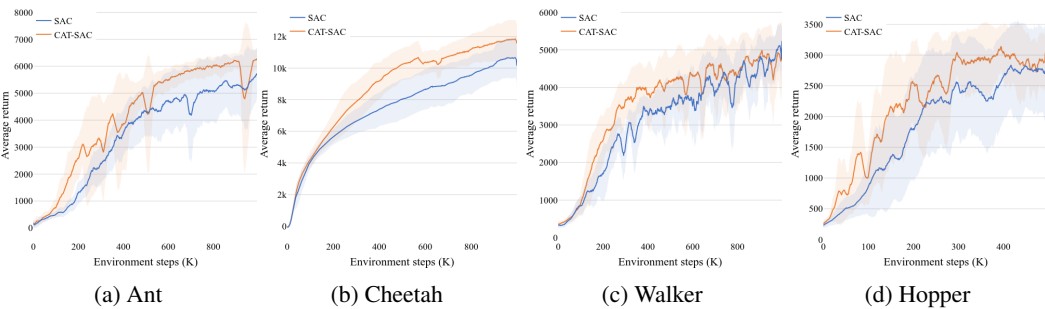

(a) Ant      (b) Cheetah      (c) Walker      (d) Hopper

Figure 1: Evaluation curves of SAC (Haarnoja et al., 2018b) and CAT-SAC on (a) Ant, (b) Cheetah, (c) Walker, and (d) Hopper environments. The solid line and shaded regions represent the mean and the standard deviation across six runs, respectively.

states from the replay buffer, and then randomly fuses any two different states with different weights. Formally, $s' = s_1 + \epsilon \times (s_2 - s_1)$, where $s_1, s_2 \sim \mathcal{D}$, $s_1 \neq s_2$ and $\epsilon \sim \mathcal{U}[0, 1.5]^{|s|}$. $|s|$ is the dimension of state feature and $\mathcal{U}[0, 1.5]^{|s|}$ is the $|s|$-dimensional uniform distribution with range $[0, 1.5]$.[2] By doing so, the synthesized 'unvisited' states still obey the similar statistical characteristics as the visited states [3]. By pushing the prediction errors of these self-constructed states above the curiosity margin, X-RND significantly improves the performance of the curiosity model with feature inputs, as demonstrated in the next section.

## 5 EXPERIMENTS

We begin by demonstrating the effectiveness of the curiosity-aware exploration strategy on difficult continuous control OpenAI Gym environments (Brockman et al., 2016) in Sec. 5.1 against current advanced model-based / model-free baselines. To better understand the benefit of the proposed curiosity model for feature inputs, we design a minimal grid-world environment to provide a concise and straightforward comparison between X-RND and RND in Sec. 5.2. After that, we ablate different choices of the important components and hyperparameters of CAT-SAC in Sec. 5.3.

### 5.1 MUJOCO RESULTS

CAT-SAC is evaluated on four continuous control tasks from OpenAI Gym (Brockman et al., 2016), i.e., Ant, Walker, Cheetah as well as Hopper. Following the same experiment settings of SUN-RISE (Lee et al., 2020), we compare CAT-SAC with METRPO (Kurutach et al., 2018), a combination of TRPO (Schulman et al., 2015) and ensembles of dynamics models; PETS (Chua et al., 2018), an advanced model-based (MB) RL method based on ensembles of dynamics models; POPLIN-P (Wang & Ba, 2019), a state-of-the-art MBRL method; POPLIN-A (Wang & Ba, 2019), a variant of POPLIN-P by injecting noise to the action space. We also compare CAT-SAC with three state-of-the-art model-free RL methods, TD3 (Fujimoto et al., 2018), SAC (Haarnoja et al., 2018b) and SUNRISE (Lee et al., 2020). For our method, we focus on the comparison between SAC and CAT-SAC. The whole algorithm for CAT-SAC is described in Alg. 1 in the Appendix. Following the experiment setups in SUNRISE, the curves and results in our experiments are calculated after 200K timesteps. Each experiment is conducted four times with the mean and the standard deviation reported. More experimental details are presented in the Appendix.

As shown in Tab. 1, CAT-SAC outperforms all of the baselines on all tasks. As for the performance on Cheetah and Walker, CAT-SAC makes a breakthrough that exceeding 6000 and 2000 on average, respectively, within 200K environment frames. On tasks Ant and Hopper, we observe that the performance of SAC and CAT-SAC almost coincides at the early stage, however, CAT-SAC gets

---

[2]With upper bound larger than 1, the value of the synthesized state can be beyond the range of the visited states.

[3]Although a few synthesized states possibly exist in the visited states, it would not have a great impact since the visited states are much more likely to be sampled during training so that their prediction error would be pulled back in expectation.

Table 1: Performance on OpenAI Gym at 200K timesteps. The results show the mean and standard deviation across six runs Henderson et al. (2017). For baselines, we report the best number in SUNRISE (Lee et al., 2020). The best results are highlighted in bold.

|  | Cheetah | Walker | Hopper | Ant |
|---|---|---|---|---|
| METRPO (2018) | $2283.7 \pm 900.4$ | $-1609.3 \pm 657.5$ | $1272.5 \pm 500.9$ | $282.2 \pm 18$ |
| PETS (2018) | $2288.4 \pm 1019.0$ | $282.5 \pm 501.6$ | $114.9 \pm 621.0$ | $1165.5 \pm 226.9$ |
| POPLIN-A (2019) | $1562.8 \pm 1136.7$ | $-105.0 \pm 249.8$ | $202.5 \pm 962.5$ | $1148.4 \pm 438.3$ |
| POPLIN-P (2019) | $4235.0 \pm 1133.0$ | $597.0 \pm 478.8$ | $2055.2 \pm 613.8$ | $2330.1 \pm 320.9$ |
| TD3 (2018) | $3015.7 \pm 969.8$ | $-516.4 \pm 812.2$ | $1816.6 \pm 994.8$ | $870.1 \pm 283.8$ |
| SUNRISE (2020) | $5370.6 \pm 483.1$ | $1926.2 \pm 694.8$ | $2601.9 \pm 306.5$ | $1627.0 \pm 292.7$ |
| SAC (2018b) | $5470.9 \pm 600.8$ | $1419.3 \pm 364.8$ | $1341.2 \pm 694.9$ | $1055.3 \pm 292.5$ |
| CAT-SAC | $\mathbf{6159.7 \pm 865.6}$ | $\mathbf{2666.2 \pm 633.0}$ | $\mathbf{3011.1 \pm 274.1}$ | $\mathbf{2444.8 \pm 856.3}$ |

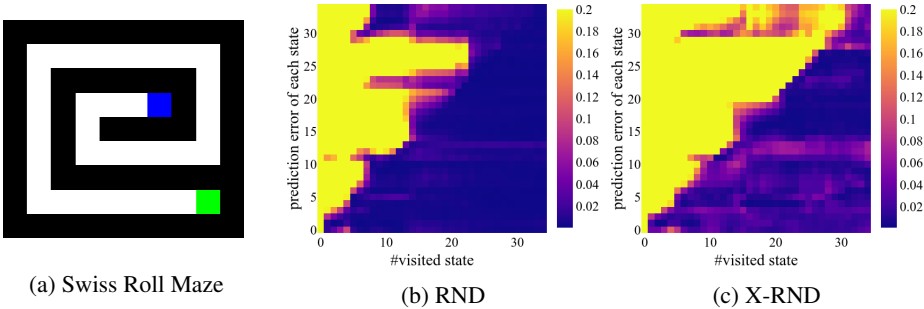

(a) Swiss Roll Maze         (b) RND         (c) X-RND

Figure 2: Demonstration on (a) the Swiss Roll maze environment, (b) prediction error of RND (c) prediction error of X-RND. The agent starts from the blue block and moves to the green block. (b)(c) shows the state prediction errors of all movable blocks (y-axis) given the number of visited blocks (x-axis) to the agent.

away from the flat slope quickly and improves faster, which implies CAT-SAC is more capable of exploiting acquired knowledge and execute an efficient exploration. In comparison with all baselines, CAT-SAC is more compute-efficient since it does not require any ensemble of dynamics and planning during the evaluation phase.

## 5.2 X-RND VERSUS RND WITH FEATURE INPUTS

We present a simple example to highlight the effectiveness and robustness of the proposed curiosity model X-RND for feature input. We introduce a grid-world environment, *Swiss Roll* maze, as depicted in Fig. 2a. The agent is moving from the blue block to the green block. Every time the agent arrives at a new block, both RND and X-RND train on features (i.e., position) of all visited blocks. Since we aim to analyze the problem of RND with feature inputs and the advantage of X-RND, we simply adopt the agent with the optimal policy for moving. We record the curiosity values of all movable blocks of both methods w.r.t. the amount of unique visited states and plots them in Fig. 2b and Fig. 2c, respectively. More experimental details are presented in the Appendix.

In Fig. 2b and Fig. 2c, values of each row present the curiosity values of all visited / unvisited. The curiosity values are clipped within the curiosity margin ($m = 0.2$ for this toy experiment). Ideally, values in the upper triangle of the figure should be closed to 0, while values in the lower triangle should be closed to the curiosity margin. From Fig. 2b and Fig. 2c, we can observe that, when the agent visits 10 unique states (index 10 at x-axis), RND has faultily regarded the distant and unvisited states (index from 30 to the last at y-axis) as visited states and produces low prediction errors for them; while X-RND successfully sustains the curiosity for these states. A more appealing property of X-RND is that X-RND is more like to form a sharp contrast between the explored and unexplored areas (with a clear diagonal). It is more important to have accurate curiosity prediction for the near unvisited states than keeping the curiosity about the very distant state high, since those near states are more likely to be visited. This phenomenon implies the synthesized states are likely to fall in the border between the explored and unexplored.

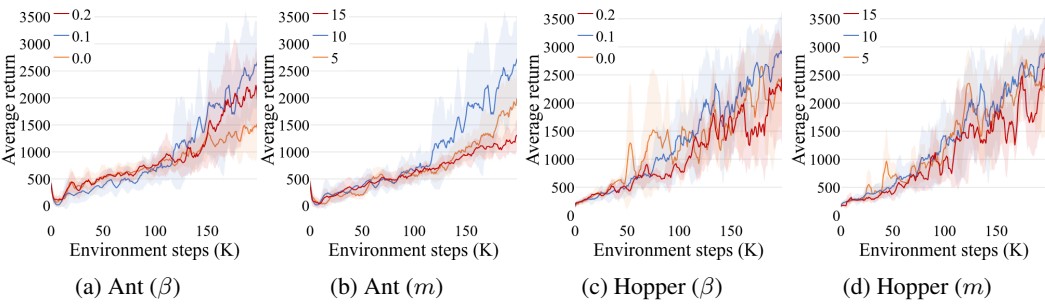

Figure 3: Evaluation curves of CAT-SAC with different values of hyperparameters $\beta$ and $m$. The solid line and shaded regions represent the mean and the standard deviation across four runs.

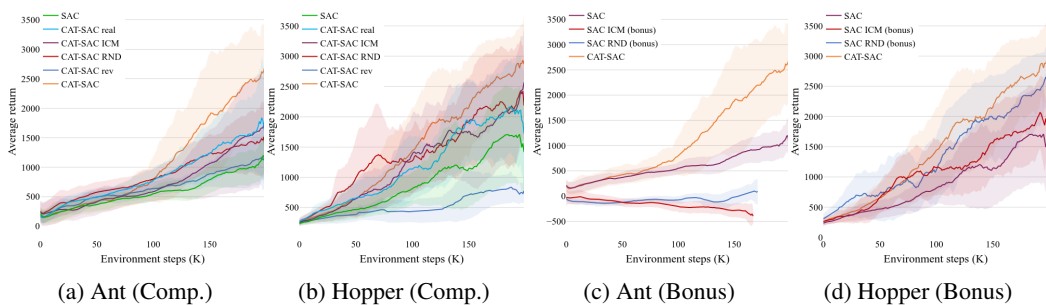

Figure 4: Evaluation curves for CAT-SAC on (a) Ant and (b) Hopper tasks. The solid line and shaded regions represent the mean and standard deviation, respectively, across four runs.

## 5.3 ABLATION STUDY

In this section, we ablate the effects of different values of the hyperparameters as well as different components of CAT-SAC on Ant and Hopper, which should be representative of all tasks.

**Effects of different hyperparameters.** We analyze the effects of different values of the hyperparameters CAT-SAC, i.e., the curiosity margin $m$ and the importance weight $\beta$ in Equ.(10). Notice that when $\beta$ equals 0, the X-RND in SAC is the same as RND. As depicted in Fig. 3a and Fig. 3c, CAT-SAC with positive $\beta$ achieves better performance on both tasks, which means that a better curiosity model X-RND indeed help improve the performance of CAT-SAC. However, we also observe that the larger $\beta$ is not necessary to bring more improvement. We suppose the reason is that a large importance weight on the contrastive loss may cause the RND loss for familiar states difficult to decrease. As for the choice of the curiosity margin $m$, we found that CAT-SAC is more sensitive to different values of $m$ on task Ant (Fig. 3b). The reason might also is that $m$ determines the range of the curiosity and therefore influence the distribution of $c(s)$, which is the critical component in CAT-SAC. But overall, with different values of $\beta$ and $m$, CAT-SAC still outperforms SAC, which implies that our method consistently gains improvement.

**Individual effect of each component.** In our full CAT-SAC, curiosity from X-RND is discretized into integer and normalized to have zero mean, and then added to the original target entropy. To evaluate the effect of each component, we conduct experiment on i) *real*, which does not discretize the curiosity 4.2; ii) *RND/ICM*, which use RND/ICM to predict curiosity; (iii) *rev*, which subtracts the curiosity from the target effect. Among these settings, only *rev* is contrary to our motivation, that's, the agent should explore more in unfamiliar states. Fig. 4a and Fig. 4b show the performance of CAT-SAC under different settings. From the results, we observe that turning off any component would decrease the performance of the full version. And except for *rev*, CAT-SAC with different component turning off still outperforms the baselines with a sharp margin. It implies that our motivation actually helpful in developing a better exploration-exploitation trade-off strategy. By utilizing all the proposed techniques, we obtain the best performance in all environments. From this comparison, we see that all proposed components have a positive effect on improving the sample-efficiency of CAT-SAC. Besides, we also conducted experiments that only augments SAC with intrinsic bonus, i.e., RND and ICM, to understand the performance of these methods. From the results shown in Fig. 4c and Fig. 4d, we found that the performance of either RND or ICM is diverse on different

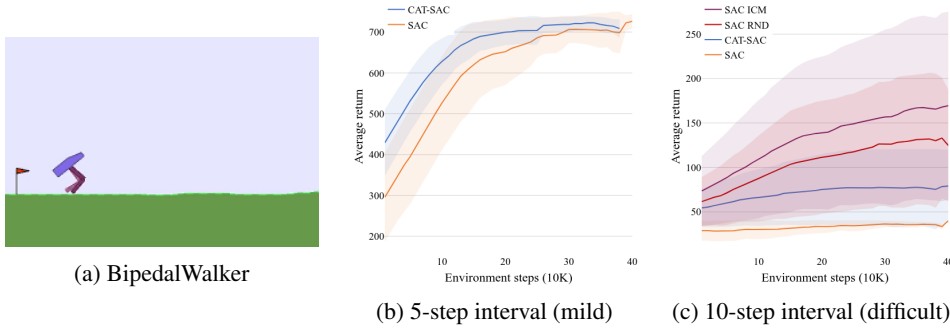

(a) BipedalWalker        (b) 5-step interval (mild)        (c) 10-step interval (difficult)

Figure 5: Evaluation curves for different approaches on (a) 5-step interval (mild) and (b) 10-step interval (difficult) tasks. The solid line and shaded regions represent the mean and standard deviation, respectively, across six runs.

tasks. Especially on task Ant, these methods seem to lead the policy to improve in the opposite direction, hampering the sample efficiency.

### 5.4 THE PERFORMANCE UNDER SPARSE-REWARD SETTINGS

On sparse-reward tasks, since the agent could not receive immediate feedback from the environment, they usually stuck in a local optimal. Intrinsic bonuses (e.g. RND (Burda et al., 2018b) and ICM (Pathak et al., 2017)) adjust the target of the agent by augmenting the environment reward with intrinsic prediction errors to encourage to form a meaningful and directional exploration behavior. Different from these methods, our methods apply curiosity in adjusting the variance of the policy, which still obeys the original target of SAC. Thus, similar to SAC, our method is not expected to solve sparse-reward tasks. Nevertheless, it is still interesting to see whether tuning the action entropy at different states w.r.t. curiosity can still gain improvement on sparse-reward tasks.

To this end, we adopt a classical Box2d (Catto, 2011) task, BipedalWalker (Fig. 5a), which is required to learn an agent to move ahead as far as it can. In our experiments, we only reward the agent if the agent reaches the anchor locations. The interval distance between the two closest anchor locations determines the difficulty of the task. In the revised BipedalWalker, the agent is required to learn to move without any feedback on its posture, and only when reaching the anchor locations will the agent be rewarded. For more environment and training details, please refer to Appendix.

In our experiments, we design two tasks with different intervals. They are 1) 5-step interval: a mild exploration task with 5-step-length interval; 2) 10-step interval: a difficult exploration task with 10-step-length interval. From Fig. 5b, we observe that on the mild task, both SAC and CAT-SAC can obtain the optimal effortlessly and CAT-SAC improves faster. However, as we increase the interval to 10 steps, both undirected exploration methods SAC and CAT-SAC perform poorly. However, CAT-SAC can sometimes get rid of the local optimal and achieves a better performance in expectation. On this difficult task, we also conduct experiments with directed exploration methods, i.e. RND and ICM. As expected, these methods achieve significantly better performances as shown in Fig. 5c. While, as discussed in the last section, introducing exploration biases like RND and ICM is not necessary to improve the sample efficiency, especially on dense-reward tasks.

## 6 CONCLUSION

In this paper, we present CAT-SAC, a simple framework unified curiosity and entropy maximization RL. In particular, CAT-SAC encourages a large entropy in unfamiliar states and a small entropy in familiar states for a better trade-off between exploration and exploitation. To model curiosity for feature input, we also propose a new prediction-based model, X-RND. Our experiments show that the full CAT-SAC consistently improves the performances of SAC and outperforms state-of-the-art RL algorithms in complex control tasks.

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

# A CAT-SAC: EXPERIMENTAL SETUPS

**Environments.** We evaluate the performance of CAT-SAC on four complex environments from OpenAI Gym (Brockman et al., 2016), i.e., Ant, Cheetah, Walker, and Hopper. Observation in these tasks is in the form of features.

**Training details.** We consider a combination of SAC and CAT-SAC using the publicly released implementation repository (https://github.com/thu-ml/tianshou) with slight modifications on hyperparameters and architectures in order to match the implementation of RLKit (https://github.com/vitchyr/rlkit). The complete algorithm is depicted in Alg. 1. The detailed hyperparameters used in CAT-SAC is presented in Tab. 2

---

**Algorithm 1** Soft Actor-Critic with Curiosity-Aware Entropy Temperature

---

1: Input: initial policy parameters $\phi$, Q-function parameters $\theta_1$, $\theta_2$, empty replay buffer $\mathcal{D}$
2: Set target parameters equal to main parameters $\theta_{\text{targ},1} \leftarrow \theta_1$, $\theta_{\text{targ},2} \leftarrow \theta_2$
3: **repeat**
4:     Observe state $s_t$ and select action $a_t \sim \pi_\phi(\cdot|s_t)$
5:     Execute $a_t$ in the environment
6:     Observe next state $s_{t+1}$, reward $r_t$, and done signal $d_t$ to indicate whether $s_{t+1}$ is terminal
7:     Store $(s_t, a_t, r_t, s_{t+1}, d_t)$ in replay buffer $\mathcal{D}$
8:     **if** $d_t$ is true **then**
9:         Reset environment state.
10:         Sample set of states $\{s\}$ from $\mathcal{D}$ and sythesize a new set of states $\{s'\}$
11:         Update $f_\omega$ by optimizing Equ.(10)
12:         Calculate the new mean $\mu$ and standard deviation $\sigma$ of $\{c(s)\}$
13:     **end if**
14:     **if** it's time to update **then**
15:         Randomly sample a batch of transitions, $B = \{(s_t, a_t, r_t, s_{t+1}, d)\}$ from $\mathcal{D}$
16:         Calculate $c(s_t)$ and $\alpha_\delta(s_t)$ for the rest optimizations
17:         Update Q-functions by one step of gradient descent using $\nabla_{\theta_i} \frac{1}{|B|} \mathcal{L}_{\text{critic}}(\theta_i)$ for i=1,2
18:         Update policy by one step of gradient ascent using $\nabla_\phi \frac{1}{|B|} \sum_{s \in B} \mathcal{L}_{\text{actor}}(\phi)$
19:         Update target networks with $\theta_{\text{targ},i} \leftarrow \rho\theta_{\text{targ},i} + (1-\rho)\theta_i$ for i=1,2
20:         Construct curiosity augmented target entropy $h(s_t)$ by Equ.(7)
21:         Update $\alpha_\delta$ by one step of gradient descent using $\nabla_\delta \frac{1}{|B|} \mathcal{L}(\delta)$
22:     **end if**
23: **until** convergence

---

**Details of the instance-level entropy temperature.** In order to automatically tune the entropy temperature in a similar way as (Haarnoja et al., 2018b), we do not adopt deep neural networks for the instance-level entropy temperature. Instead, we initialize a list of scalar variables. Each variable is assigned to a integer index, ranging from $-|$target entropy$|$ to $|$target entropy$|$. Given the discrete curiosity input, we match it with the variable with the closest index, and then the same optimization process in (Haarnoja et al., 2018b) is conducted to optimize these variables. As for the input under the *real* setting in Sec. 5.3, we match it with the two variables with the closest indexes and calculate the weights of these variables so that the sum of the weighted indexes assembles the curiosity value. Then the sum of the weighted variables is treated as the entropy temperature. The implementations of different approaches as depicted in Fig. 6.

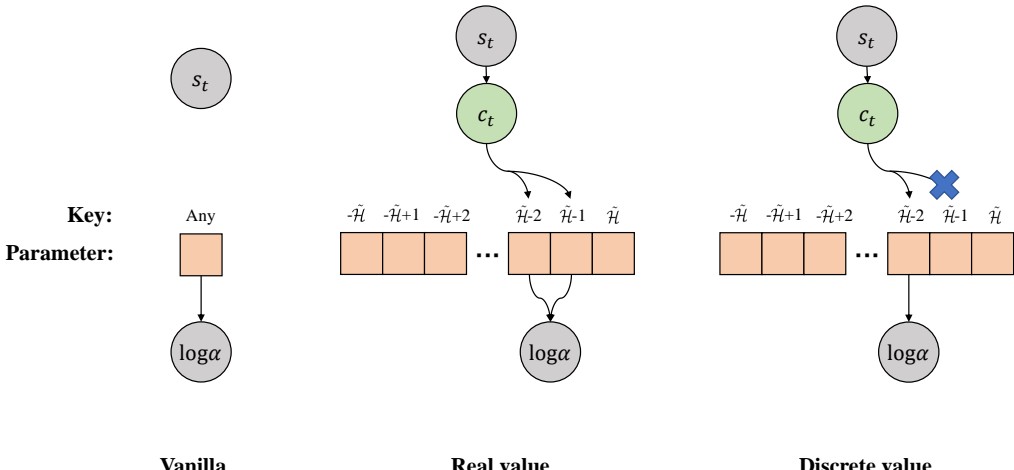

Figure 6: The sketches of $\log \alpha$. The left one is the implementation of vanilla SAC (Haarnoja et al., 2018b). And the others are proposed in our paper to model instance-level entropy temperature. The middle one explain how we handle real-value curiosity. And the right one first rounds the curiosity into the closest integer and then choosing the matched entropy temperature.

Table 2: Values of hyperparameters used in this paper.

| Hyperparameter | Value | Hyperparameter | Value |
|---|---|---|---|
| #training frames | 2e7 | #frames between evaluations | 1e3 |
| Backbone | MLP | Non-linearity | ReLU |
| Backbone ($g$) | Vector | (X-)RND feature size | 128 |
| Hidden layers | 2 | Optimizer | Adam |
| Hidden layers ((X-)RND $f$) | 4 | Critic target update freq | 1 |
| Hidden units | 256 | Learning rate | 3e-4 |
| Batch size | 256 | Discount $\gamma$ | 0.99 |
| Initial temperature | 1 | Curioisty margin $m$ | 10 |
| #batch for update (X-)RND | 100 | Importance weight $\beta$ | 0.1 |
| Polyak factor $\tau$ | 0.005 | Target entropy | $-\mid$ action space $\mid$ |
| Replay buffer size | 1e6 | Evaluation episodes | 10 |
| Hidden layer (S.R.) | 1 | Hidden units (S.R.) | 16 |
| Importance weight $\beta$ (S.R.) | 0.5 | (X-)RND feature size (S.R.) | 16 |
| Curiosity margin $m$ (S.R.) | 0.2 | Batch size (S.R.) | 64 |
| Hidden layer (Bipedal.) | 2 | Hidden units (Bipedal.) | 16 |
| Importance weight(Bipedal.) $\beta$ | 0.1 | (X-)RND feature size (Bipedal.) | 128 |
| Curiosity margin (Bipedal.) $m$ | 0.3 | Batch size (Bipedal.) | 128 |

# B  SWISS ROLL MAZE: EXPERIMENTAL SETUPS

**Environment.** As depicted in Sec. 5.2. In this environment, only position features are provided.

**Training details.** Most of the hyperparameters take the same values as CAT-SAC, presented in Tab. 2. For some hyperparameters with different values, we tag them with '(S.R.)' in the table. The visualization results of the prediction error of RND and X-RND are presented in Fig. 7 and Fig. 8, respectively. From these figures, we can observe the X-RND can maintain the curiosity about distant states with feature input.

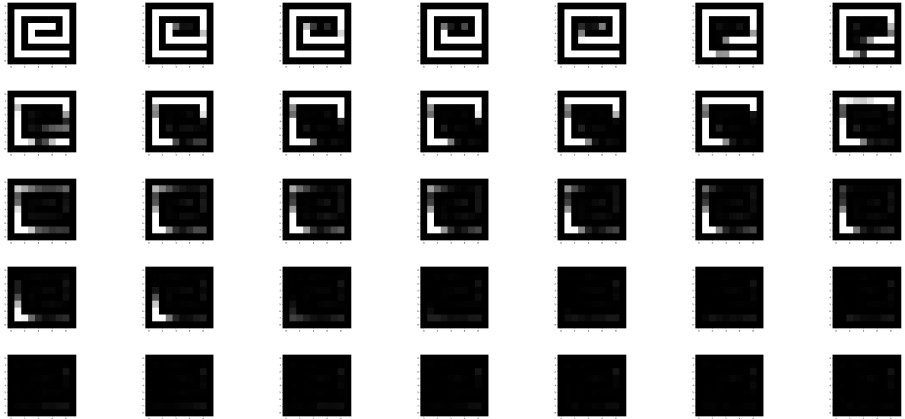

Figure 7: The prediction errors of all blocks of RND. These figures are obtained by using RND to predict the curiosity about all states after training with different numbers of visited states. From left to right and from top to bottom, the number of visited states is increasing.

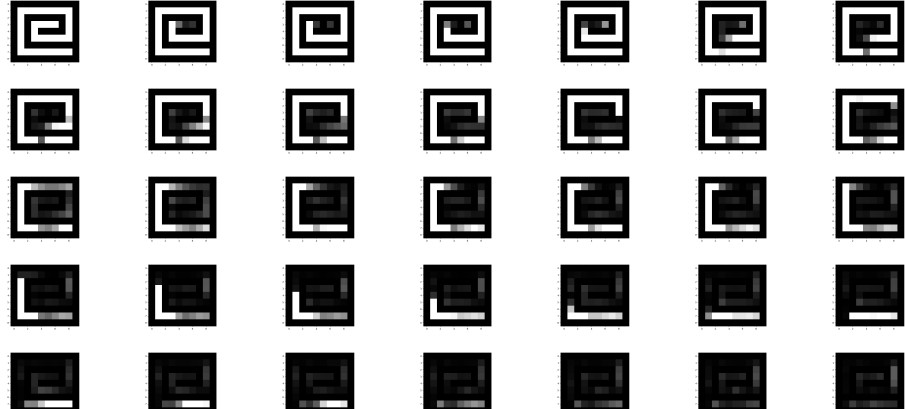

Figure 8: The prediction errors of all blocks of X-RND. These figures are obtained by using X-RND to predict the curiosity about all states after training with different numbers of visited states. From left to right and from top to bottom, the number of visited states is increasing.

## C BipedalWalker: Experimental Setups

**Environment.** We adopted the open-released BipedalWalker and revised it to the sparse-reward task by removing its original reward function. The revised version rewards the agent if the agent reach predefined anchor locations. These locations are placed along the track evenly with fixed interval. Since there are no posture reward, an agent is required to explore how to walk to these anchor locations. We also augmented the original state feature with the hull's location (x, y) of the agent.

**Training details.** Most of the hyperparameters take the same values as CAT-SAC, presented in Tab. 2. For some hyperparameters with different values, we tag them with '(Bipedal.)' in the table.

