# OpenReview forum: "CAT-SAC: Soft Actor-Critic with Curiosity-Aware Entropy Temperature"
_ICLR.cc/2021/Conference — Reject_

### Official Review · AnonReviewer1 · 2020-10-28
**Intuitvely reasonable and empirically beneficial exploration strategy**

**Rating:** 6
**Confidence:** 3

**Review:**

**Contribution**: For better exploration, the authors propose to use curiosity to set state dependent target entropies with SAC, with the goal of inducing more diverse behavior at unfamiliar states. They use RND to provide a curiosity score, which after normalizing, is used to adjust the state dependent target entropy. Due to RND performing poorly as a curiosity measure with state-based representations instead of image based, introduce a variant X-RND which additionally uses a contrastive loss to improve the curiosity mechanism. They demonstrate benefits over regular SAC on standard Mujoco Gym benchmarks.

**Prior work in exploration**: While increasing policy entropy in regions of low  confidence makes intuitive sense, it is not actually clear simply being noisier in the face of uncertainty actually leads to good exploration (for example https://arxiv.org/abs/1306.0940 argues that these "dithering" style exploration is inefficient). It would be good to add additional comparisons to other methods for augmenting SAC to perform better exploration. For example, OAC https://arxiv.org/abs/1910.12807 learns upper confidence bounds on Q-values and uses the optimistic Q values for an exploration policy. It could be interesting to also explore how well CAT-SAC compares to just using the (X-)RND curiosity score directly as a bonus for exploration. In general, the related work is lacking in discussion of classical RL exploration methods like UCB-style bonuses and posterior sampling.

**Conclusion**: Overall, I like the work. It appears technically sound, and presents a fairly simple and intuitive way to adjust exploration with SAC to better handle unfamiliar states (as far as any undirected exploration methods do at least). I would like to see a few more comparisons against other exploration techniques in deep RL, and perhaps some experiments ont asks outside of the standard gym benchmarks that focus more on the exploration problem itself rather than control (for example some sparse reward tasks).

Fairly minor points:
LaTeX error: There are several instances of log in math mode that should use \log instead.
I would also like to see extended learning curves of CAT-SAC vs SAC, particularly until we see the performance of each method saturate. It would be interesting to see if the better exploration from CAT-SAC allows it to converge to higher performing policies as well as learning faster.

---

> ### Author Response · Authors · 2020-11-23
> **Extend related work and add more ablation studies**
>
> Thank you for your helpful clarification of our settings (an undirected exploration strategy) and constructive suggestion for our work. We express our sincere apologies to the reviewer for the late reply because the extended experiments have cost us a long time.
>
> The idea of increasing entropy at unfamiliar states has been studied in [1], which demonstrates the effectiveness of adaptively tuning the epsilon at different states w.r.t. familiarity of states. And in our ablation study, we also found that if we take an opposite strategy, that’s, explore more at familiar states and exploit more at unfamiliar states, the performance of the agent becomes worse, as depicted in Fig.4(a) and Fig.4(b). Similar ideas are also applied to uncertainty-based exploration strategies, e.g. upper-confidence-bound exploration. As for the comparison with UCB methods, one of the baselines in our paper, SUNRISE, is based on UCB for exploration and achieves superior performance. As reminded by the reviewer, we have added more comprehensive related works in our revised manuscript.
>
> According to the suggestion, we have added a series of new experiments to answer:
>
> __Q:  How about adding an extra exploration bonus, e.g. RND, ICM, to SAC?__
>
> We have added the experiments with RND and ICM to SAC in Sec.5.3. From the results in Fig.4(c) and Fig.4(d), we can see that for different tasks, the performances of RND and ICM are diverse. Especially on tasks Ant, both methods tend to teach an agent in the opposite direction. It implies that introducing directional exploration signals might lead the agent to improve in the opposite direction.
>
> __Q: How about applying SAC / CAT-SAC on sparse reward tasks?__
>
> To understand the performance of baselines under sparse reward settings, we adopt the BipedalWalker task and revise it to be a sparse reward task, as elaborated in Sec.5.4. From the results in Fig.5(a), we can see that when the difficulty of exploration is mild, both SAC and CAT-SAC can obtain the optimal effortlessly while CAT-SAC improves faster. However, when it comes to the harder setting, as depicted in Fig.5(b), SAC fails to obtain improvement while CAT-SAC sometimes can escape from the local optimal, resulting in a better average performance.  Within expectation, both SAC/CAT-SAC perform poorly in comparison with directional exploration approaches, i.e. RND and ICM.
>
> __Q: How about running longer?__
>
> In our revised manuscript, we extend our major experiment in Sec.5.1 by training each experiment longer until it converges or the maximum environment step 1e6 is reached. On all of the tasks, our CAT-SAC consistently improves faster. Especially for tasks Ant and Hopper, we found that our methods converge within 1e6. However, observed from the converged results, it seems like CAT-SAC does not improve the upper bound of the performance.
>
>
>
> [1] Tokic M. Adaptive ε-greedy exploration in reinforcement learning based on value differences[C]//Annual Conference on Artificial Intelligence. Springer, Berlin, Heidelberg, 2010.

---

### Official Review · AnonReviewer3 · 2020-10-28
**Interesting way of incorporating curiosity but requires clarification and more experiments**

**Rating:** 4
**Confidence:** 4

**Review:**

Summary:

This work proposes to incorporate curiosity into the entropy temperature of Soft Actor-Critic, and applies a modified version of Random Network Distillation as their curiosity model. Its key insight is that the entropy temperature in Soft Actor-Critic should encourage the agent to explore unfamiliar states more and familiar states less, rather than globally encourage some expected target entropy. To this end, they make the target entropy and temperature both state-dependent, using a curiosity model based on Random Network Distillation.



Strengths:

- The insight is described well and motivates the need for a curiosity-aware entropy temperature for further sample-efficiency.

- The paper demonstrates good experimental results across four OpenAI Gym tasks evaluated after 200K environment steps. There is also a thorough ablation study to illustrate the importance of each component of the approach.



Weaknesses:

- The justification of the instance-level entropy temperature in Section 4.2 could be clearer. As is, it’s unclear why the instance-level entropy is not redundant after the curiosity augmented target entropy, and vice versa. It’s also not well justified why the curiosity value is rounded and then re-scaled to form the entropy temperature.

- While the method achieves good results on the standard tasks, I am still curious about its performance on pure exploration tasks, and hard exploration tasks such as sparse-reward settings. In the former, it would be great to quantitatively measure the unique states visited and to provide a comparison to RND.

- While the modification to RND is intended to correct for curiosity at unseen states, the selection of unseen states is based on a heuristic: unseen states are created as combinations of pairs of seen states.



Recommendation:

I am recommending to reject this paper. I think parts of the method can be more clearly explained and justified. There is also potential to better highlight the strengths of the proposed method in the experimental results, i.e., by including pure/hard exploration tasks.



Questions:

- The learning curves in Figure 1 evaluate SAC and CAT-SAC for 200K environment steps. I’m curious about the behavior after more steps & at convergence, and how many steps are necessary for convergence.

- Are beta and m tuned for each task?

- What does “the instance-level entropy temperature may weaken the connection across states” in Section 4.2 mean?



Updates after Reading Authors' Response:

Thank you for the detailed clarification and the new results. From the response, it seems that the proposed modifications to SAC do not and are not meant to solve the exploration problem in sparse-reward tasks. Instead, its aim is to improve sample efficiency on standard dense-reward tasks. The impact of the work then feels quite limited: the proposed modifications are specific to SAC and do not meaningfully improve performance on the sparse-reward task. For these reasons, I will keep my original evaluation.

---

> ### Author Response · Authors · 2020-11-23
> **Reorganize Sec.4.2 and provide more explanation as well as more ablation studies (PART 1)**
>
> Thank you for your constructive suggestion and comprehensive comments on our approach. We express our sincere apologies to the reviewer for the late reply because the extended experiments have cost us a long time.
>
> We have rewritten Sect. 4.2 to make it more clear.
>
> The augmented target entropy will have an average impact on Equ.(8) if there is a global entropy temperature. If there is a global entropy temperature, one can extract the entropy temperature outside of the expected term. And then Equ.(8) will be equal to Equ.(5) since the curiosity value is rescaled into zero-mean to maintain the original target entropy in expectation.
>
> The reason why we rounded the zero-mean curiosity into an integer is to make the states with close curiosities to share the same target entropy term. As explained in Sec.4.2 in our revised manuscript, assigning different entropy temperature for each state independently makes the optimization of Equ.(8) trivial: the entropy temperature becomes either extremely high or extremely low according to the sign of $\log\pi_\phi(a_t|s_t) + h(s_t)$.  It further over-weights $\log\pi_\phi(a_t|s_t)$ of Equ.(4), which influences back the optimization of Equ.(8). The consequence of these optimization processes is that $\log\pi_\phi(a_t|s_t)$ is optimized to approach $h(s_t)$ rapidly. As mentioned by the authors of SAC[1], forcing the entropy to be a target entropy will hamper the flexibility of the policy. To this, we rounded the zero-mean curiosity into integers to make the states with close curiosities to share the same target entropy term.
>
> The combination of two visited states to generate out-of-manifold samples has been well-studied in the field of computer vision, named as mixup [2]. Even though in our paper, we use this method on feature inputs, it has a chance to generate a visited state as has been stated in original manuscripts (footnote 3). But this probability is much smaller than the visited state being sampled from the replay buffer as the positive training samples. Thus it should not bring about much impact. As has been visually demonstrated in our Swiss Roll experiments, synthesizing unvisited states in this manner are likely to result in a sharp contrast between the explored and unexplored areas. This implies that such an operation can generate unvisited states around the border between the explored and unexplored areas.
>
> Beta and m for each task are fixed as 0.1 and 10 on MuJoCo, respectively. For different types of tasks (i.e., MuJoCo, BipedalWalker), beta is set as 0.1 empirically. And as for the value of m, in practice, we first trained the RND model after several iterations to obtain an approximate upper bound of the curiosity. And then we set the m as the approximate upper bound. As ablated in Sec.5.3, the proposed method can still obtain improvement with different beta and m. Thus, one could obtain improvement by roughly selecting the value of beta and m via the strategy mentioned above, and can further tune these parameters to achieve better performance.
>
> As for evaluating our method under sparse reward settings, the improve the exploration of SAC on continuous control tasks. The improvement of our methods is not achieved by introducing directed or temporal exploration bonuses like RND[1] and ICM[2] to the policy (as clarified by the reviewer1). Different from ours, these methods augment the environment rewards with extrinsic exploration bonuses to encourage the policy to transition to unknown states, which has been demonstrated to significantly improve the performance on sparse reward tasks. However, the introduced exploration biases sometimes might harm the efficiency of SAC on dense reward tasks. Our approach aims to improve the original SAC by considering more on the variance of policy under its original theoretical framework. Therefore, like SAC, our improved version is not expected to solve the sparse reward exploration problem, acknowledged by the author of SAC([Link](https://github.com/haarnoja/sac/issues/5)). Nevertheless, it seems necessary to conduct experiments on sparse-reward tasks to understand our approach from a more comprehensive perspective.
>
>
>
> [1] Haarnoja T, Zhou A, Hartikainen K, et al. Soft actor-critic algorithms and applications[J]. arXiv preprint arXiv:1812.05905, 2018.
>
> [2] Guo H, Mao Y, Zhang R. Mixup as locally linear out-of-manifold regularization[C]//Proceedings of the AAAI Conference on Artificial Intelligence. 2019, 33: 3714-3722.

---

> ### Author Response · Authors · 2020-11-23
> **Reorganize Sec.4.2 and provide more explanation as well as more ablation studies (PART 2)**
>
>
> In conclusion, in our revised manuscript, we have conducted a series of experiment to answer:
>
> __Q:  How about adding an extra exploration bonus, e.g. RND, ICM, to SAC?__
>
> We have added the experiments with RND and ICM to SAC in Sec.5.3. From the results in Fig.4(c) and Fig.4(d), we can see that for different tasks, the performances of RND and ICM are diverse. Especially on tasks Ant, both methods have the tendency to teach an agent in the opposite direction. It implies that introducing directional exploration signals might lead the agent to improve in the opposite direction.
>
> __Q: How about applying SAC / CAT-SAC on sparse reward tasks?__
>
> To understand the performance of baselines under sparse reward settings, we adopt the BipedalWalker task and revise it to be a sparse reward task, as elaborated in Sec.5.4. From the results in Fig.5(a), we can see that when the difficulty of exploration is mild, both SAC and CAT-SAC can obtain the optimal effortlessly while CAT-SAC improves faster. However, when it comes to the harder setting, as depicted in Fig.5(b), SAC fails to obtain improvement while CAT-SAC sometimes can escape from the local optimal, resulting in a better average performance.  Within expectation, both SAC/CAT-SAC perform poorly in comparison with directional exploration approaches, i.e. RND and ICM.
>
> __Q: How about running the experiment longer?__
>
> In our revised manuscript, we extend our major experiment in Sec.5.1 by training each experiment longer until it converges or the maximum environment step 1e6 is reached. On all of the tasks, our CAT-SAC consistently improves faster. Especially for tasks Ant and Hopper, we found that our methods converge within 1e6. However, observed from the converged results, it seems like CAT-SAC does not improve the upper bound of the performance.

---

### Official Review · AnonReviewer2 · 2020-10-31
**Interesting idea but needs more rigorous experiments**

**Rating:** 4
**Confidence:** 3

**Review:**

This paper introduces an algorithm that augments SAC with Curiosity-Aware Temperature, to enable more efficient exploration. Previous versions of SAC had a fixed entropy temperature which had to be tuned or an automatic tuning mechanism that was not state-specific. The paper proposes that exploration can be improved if temperature is state-dependent, based on curiosity, or unfamiliarity of the state.

Authors introduce curiosity to the target temperature such that the entropy is large in unfamiliar states, promoting exploration, and small in familiar states, encouraging more exploitation. To enable this, the authors introduce three components: 1) target entropy that is augmented with curiosity, 2) curiosity and hence state based entropy, 3) X-RND that adds contrastive loss to ensure more robust computation of curiosity.

Curiosity is based on prediction error of states, using the idea from Random Distillation Network (RND) (Burda et al. 2018b). This curiosity term is normalized such that in expectation it corresponds to the original target entropy. The instance-level temperature also uses this curiosity to map states with similar level of unfamiliarity to similar temperature value. X-RND is a technique that the authors develop in order to overcome previous difficulties that RND had on feature inputs.

In their benchmark experiments they show that their method CAT-SAC shows superiority compared to SAC as well as other baseline methods. They also show results on a toy domain how X-RND can estimate curiosity more robustly than RND.

The authors have done a fair job introducing curiosity to enable better exploration by varying the target entropy at state-level. Disregarding minor grammar errors I think it is structured nicely. The idea is interesting but I would recommend reject as the experiments need to be conducted more rigorously.

Benchmark experiments are conducted only with 4 runs on Mujoco environments that are known to have high variance based on random seeds. Henderson et al. 2018 (https://arxiv.org/pdf/1709.06560.pdf) show that on HalfCheetah, the same algorithm can have significantly different performance, between two groups of 5 random seeded runs. More runs need to be conducted in order to show credible performance improvements.
Furthermore, currently CAT-SAC surpasses SAC in all four Mujoco domains at 200k steps of training, but SAC results in the original paper (Haarnoja et al. 2018b) show SAC reaching avg return of 2000 in Hopper and 3200 in Walker. This also shows that current results where SAC performs worse than CAT-SAC could have been due to variance in random seeds.

Comparison of CAT-SAC to other baseline method performance seems less appropriate too. Results are based on SUNRISE paper (Lee et al. 2020) which has not been peer-reviewed and which have also done only 4 runs each.

Lastly, demonstration of X-RND seems to show that it is able to remember states it has visited and keep curiosity for remaining unvisited states. It looks like after the state is visited once the curiosity drops from 0.2 to 0.02 immediately. I’m not sure if it would be desirable to have curiosity drop suddenly after visiting it only once. To me X-RND seems like a mechanism with a replay buffer that limits neural networks from generalizing and making it tabular-like to output low curiosity only for states it has visited. X-RND certainly performs better than RND in this toy domain but I’m not sure if it adds that much value to the overall paper, where it is about having instance-level entropy that encourages exploration.

For other minor details, I think the plots and labels in Figure 2(b) (c)  are confusing.I think the x-axis should be index of each state, and prediction error of each state the label for the colormap.

---

> ### Author Response · Authors · 2020-11-23
> **Run the experiments with more random seeds and longer**
>
> Thanks for your constructive comments and considerate suggestions to our manuscript. We express our sincere apologies to the reviewer for the late reply because the extended experiments have cost us a long time.
>
> We agree with the reviewer on the high variance of RL studied by Henderson et al. 2018 [1]. However, Henderson et al. conducted experiments on two tasks, i.e., Cheetah and Hopper, using TRPO and DDPG, instead of a more stable method, i.e. SAC. SAC has been shown as a much stable method on MuJoCo [2]. To alleviate the effect of random seeds, in our revised manuscript, we have conducted six repeated experiments between SAC and CAT-SAC across four different tasks, with their standard variance reported, as depicted in Tab.1 and Fig.1. And our method has shown consistent improvement on all of these tasks.
>
> Also mentioned by Henderson et al. [1], different codebases seem to have a non-negligible impact on the performance, therefore it is reasonable to see different performances between the original paper and ours. According to the results reported in our paper, our implemented SAC actually achieves quite close performance to the original paper. Particularly, on task Ant, our implementation converges faster. Even in comparison with the original paper, our CAT-SAC can still achieve superior performance. In our manuscript, we choose to compare with SUNRISE because it achieves state-of-the-art performance under the same settings as ours.
>
> As for curiosity modeling, the curiosity modeled by either RND or X-RND does not drop suddenly after visiting the state once. More precisely, the curiosity of (X-)RND w.r.t. a visited state gradually decreases as the state is sampled to train repeatedly. In practice, the curiosity of the state is likely to be smaller if the state is more often visited and has more chances to be sampled to train (X-)RND, as studied in [3]. Introducing the contrastive loss does not endow the X-RND with the ability to memorize states immediately. The contrastive loss is adopted to prevent the curiosity of the unvisited states from being affected by the visited and similar states. (X-)RND in the toy experiment is trained to converge whenever a new state is visited, to better demonstrate the problem of RND in modeling curiosity with feature input. Observing from this experiment, X-RND is more likely to maintain a high curiosity value for an unvisited/unfamiliar state, which is desirable to our proposed CAT-SAC. The ablation studies on MuJoCo also demonstrate the benefit of X-RND over RND.
>
> As for the plot, we have switched the axes for better reading. Thank you for your helpful reminders.
>
> [1] Henderson, P., Islam, R., Bachman, P., Pineau, J., Precup, D., & Meger, D. Deep reinforcement learning that matters. AAAI 2018.
>
> [2] Haarnoja T, Zhou A, Abbeel P, et al. Soft Actor-Critic: Off-Policy Maximum Entropy Deep Reinforcement Learning with a Stochastic Actor[C]//ICML. 2018.
>
> [3] Burda Y, Edwards H, Storkey A, et al. Exploration by random network distillation[C]//ICLR. 2018.

---

### Official Review · AnonReviewer4 · 2020-11-01

**Rating:** 4
**Confidence:** 5

**Review:**

Summary: The paper tries to improve exploration in continuous control tasks by augmenting Soft Actor-Critic (SAC) with a curiosity module that increases the target entropy for unfamiliar states and decreases the target entropy for familiar states. The curiosity module is implemented using RND combined with a contrastive loss function (X-RND). To incorporate curiosity into SAC, the entropy coefficient is learned and dependent on the prediction error (coming from X-RND) of the state. The overall method (CAT-SAC) is tested on mujoco and swiss roll maze. Additionally, the paper includes ablation studies showing the benefit of X-RND and that of discretization of curiosity.

Novelty: The main novel component is incorporating curiosity into entropy to increase the target entropy in unfamiliar states and decrease it in familiar states.

Pros: CAT-SAC does improve over SAC on mujoco domains and it helps to have X-RND component instead of vanilla RND.

Cons: The ablation studies and the evaluation feels incomplete.
(1) Rather than adding curiosity to the entropy, what if we added it to the reward function? How well that perform? That should be a baseline.
(2) RND doesn’t work that well with feature inputs. What about other notions of curiosity? How does dynamics model prediction error work in input space? There should be 2 baselines: (i) how well dynamics model prediction error work when added to reward function? (ii) how well it works when incorporated into entropy?
(3) The method should be evaluated on harder exploration tasks (eg: ant gather and swimmer gather from Variational Information Maximizing Exploration (VIME; Houthooft et. al.) paper) and compared with VIME (or other exploration algorithms eg: why does hierarchy work so well? (nachum et. al.))

Reasons for score: Weighing the above pros and cons, I vote for rejecting.

Questions: Given c(s_t)is a scalar (as it’s prediction error), is g_{\delta}(c(s_t)) just m(c(s_t)) + b?

I am happy to reconsider my score if the above concerns are addressed.

---

> ### Author Response · Authors · 2020-11-23
> **Add more ablation studies and add the sketch of the structure of the entropy temperature**
>
> Thank you for your comprehensive comments and constructive suggestion on experiments. We express our sincere apologies to the reviewer for the late reply because the extended experiments have cost us a long time.
>
> As summarized by the reviewer, our proposed method is based on Soft Actor-Critic and trying to improve the exploration of SAC on continuous control tasks. The improvement of our methods is not achieved by introducing directed or temporal exploration bonuses like RND[1] and ICM[2] to the policy (as clarified by the reviewer1). Different from ours, these methods augment the environment rewards with extrinsic exploration bonuses to encourage the policy to transition to unknown states, which has been demonstrated to significantly improve the performance on sparse reward tasks. However, the introduced exploration biases sometimes might harm the efficiency of SAC on dense reward tasks. Our approach aims to improve the original SAC by considering more on the variance of policy under its original theoretical framework. Therefore, like SAC, our improved version is not expected to solve the sparse reward exploration problem, acknowledged by the author of SAC([Link](https://github.com/haarnoja/sac/issues/5)). Nevertheless, it seems necessary to conduct experiments on sparse-reward tasks to understand our approach from a more comprehensive perspective.
>
> As for the reason why we choose the state-based unfamiliarity modeling method, i.e. RND, instead of a transition-based unfamiliarity method, e.g. ICM, it is because that the optimization target of the actor in SAC (Equ. (4) in our manuscript) is conditioned on state and its action term is sampled from the policy. Therefore, the ground-truth next state, w.r.t. the state and the sampled action, is unavailable during training, which is required by transition-based unfamiliarity methods. Even so, in our revised manuscript, we still attempt to combine ICM with our method. To do so, we adopt the dynamics prediction error based on the collected transition, ignoring the difference between the sampled action and the actual action in the collected transition.
>
> Given $c(s_t)$ is a scalar, the structure of $g_{\delta}(c(s_t))$ is a one-dimensional embedding layer as depicted in Fig.6. We design $g$ in this way to follow the same structure as the vanilla SAC.
>
> In conclusion, in our revised manuscript, we have conducted a series of experiment to answer:
>
> __Q:  How about adding extra exploration bonuses, e.g. RND and ICM, to SAC?__
>
> We have added the experiments with RND and ICM to SAC in Sec.5.3. From the results in Fig.4(c) and Fig.4(d), we can see that for different tasks, the performances of RND and ICM are diverse. Especially on tasks Ant, both methods fail to improve the policy. It implies that introducing a directed exploration bonus might lead the agent to improve in the opposite direction, resulting in a poor sample-efficiency.
>
> __Q: How about replacing RND with ICM in our proposed framework?__
> As mentioned, RND has difficulty modeling curiosity for feature input, while dynamics-based methods like ICM might alleviate this problem. We conduct experiments by simply replacing the (X-)RND predictor with the ICM module proposed. As depicted in Fig.4 (a) and Fig.4(b), both approaches actually achieve quite similar performance, which implies that ICM for feature inputs might also have the same problem as RND for feature input. This could happen when the learned dynamics is also applicable for unvisited and distant states.
>
> __Q: How about applying SAC / CAT-SAC on sparse reward tasks?__
>
> To understand the performance of baselines under sparse-reward settings, we adopt the BipedalWalker task and revise it to be a sparse-reward task, as elaborated in Sec.5.4. From the results in Fig.5(a), we can see that when the difficulty of exploration is mild, both SAC and CAT-SAC can obtain the optimal effortlessly while CAT-SAC improves faster. However, when it comes to the harder setting, as depicted in Fig.5(b), SAC fails to obtain any improvement while CAT-SAC sometimes can escape from the local optimal, resulting in a better average performance.  Within expectation, both SAC/CAT-SAC perform poorly in comparison with directed exploration approaches, i.e. RND and ICM. However, on dense-reward tasks, these directed exploration bonuses might bring about side effects as discussed above.
>
>
>
>
> [1] Burda Y, Edwards H, Storkey A, et al. Exploration by random network distillation[C]//ICLR. 2018.
>
> [2] Pathak D, Agrawal P, Efros A A, et al. Curiosity-driven exploration by self-supervised prediction[C]//Proceedings of the IEEE Conference on Computer Vision and Pattern Recognition Workshops. 2017: 16-17.

---

### Author Response · Authors · 2020-11-25
**To all reviewers**

Thank you for your constructive comments. We have adjusted our manuscript accordingly.

According to the comments from all reviewers, we concluded the two major concerns/interests of the reviewers.

__How about running longer (i.e., more than 200K)?__

__How about sparse-reward tasks?__

As for the second concern, we believe this is not the exact problem our method is addressing. The improvement of our method is not achieved by introducing directed or temporal exploration bonuses like RND[1] and ICM[2] to the policy (as clarified by the reviewer1). Different from ours, these methods augment the environment rewards with extrinsic exploration bonuses to encourage the policy to transition to unknown states, which has been demonstrated to significantly improve the performance on sparse reward tasks. However, the introduced exploration biases sometimes might harm the efficiency of SAC on dense reward tasks. Our approach aims to improve the original SAC by considering more on the variance of policy under its original theoretical framework. Therefore, like SAC, our improved version is not expected to solve the sparse reward exploration problem, acknowledged by the author of SAC[Link](https://github.com/haarnoja/sac/issues/5). Nevertheless, it seems necessary to conduct experiments on sparse-reward tasks to understand our approach from a more comprehensive perspective.

In conclusion, to address the reviewers' suggestions and concerns, we added further experimental results and made significant clarifications to the paper, while striving to keep the broad ideas and findings intact. In summary, the following changes have been made:

1) We have rewritten section 4.2 and added a model sketch in Fig.6 to improve the clarity.
2) We have rerun our major experiments for six random seeds and longer. Results are updated in Fig.1 and Tab.1.
3) We have extended our ablation studies with a) Pure RND/ICM experiments b) Replacing RND with ICM in our approach, as well as c) Adding sparse-reward experiments.


[1] Burda Y, Edwards H, Storkey A, et al. Exploration by random network distillation[C]//ICLR. 2018.

[2] Pathak D, Agrawal P, Efros A A, et al. Curiosity-driven exploration by self-supervised prediction[C]//Proceedings of the IEEE Conference on Computer Vision and Pattern Recognition Workshops. 2017: 16-17.

---

### Decision · Program_Chairs · 2021-01-07
**Final Decision**

**Decision:**

Reject

**Comment:**

The concept of increasing entropy in novel states to promote exploration appears to be quite interesting. I do appreciate this idea, and I would encourage the authors to study it further. I think the reviewers also agree with this. Unfortunately, the paper as written has a number of issues: (1) the theoretical motivation for this is quite weak -- it intuitively makes sense, but for a published paper, we need more than intuition; (2) the empirical results are not especially compelling, it seems like the authors have to kind of thread the needle in arguing that they are concerned specifically with dense reward exploration -- a less chartable view is that the method just doesn't work all that well compared to other exploration settings in problems that present a major exploration challenge. The reviewers generally found the evidence in favor of the method to be a bit questionable. Therefore, in the balance, while the paper presents what I think is a really nice idea, it still needs work to justify both theoretically and empirically. I would encourage the authors to flesh out this work more -- I think with a bit more work, it could be a really nice paper, but for now it's probably not quite ready.

A few more comments (which did not influence the decision, but I recommend addressing in the future):

- Presenting exploration and efficiency results in a table, like Table 1, is not good. It hides actual patterns in performance, especially if you are talking about exploration. To the authors' credit, it appears that this trend was started by prior work (e.g., Kimin Lee et al.), but it was bad scholarship then, and it's bad scholarship now -- it's quite easy to pick a checkpoint where a given method looks better than all the other methods (which might be why some prior work opted for this format), but it's misleading to the reader and should not be done.

- I don't agree with Reviewer 2's comments about random seeds. Certainly it's better to have more random seeds than less, but this doesn't appear out of line with the standard in the field. That said, the results in Figure 4 do look quite close, so trying an actual statistical significance test might be a good idea (again, this didn't influence my decision -- the standard in RL holds that 4-6 seeds is plenty, and we have to review work by the current standard in the field).